# Measuring the seismic risk along the Nazca-Southamerican subduction front: Shannon entropy and mutability

Eugenio E. Vogel[1,2], Felipe G. Brevis[1], Denisse Pastén[3,4], Víctor Muñoz[3],
Rodrigo A. Miranda[5,6], Abraham C.-L. Chian[7,8]

[1]Departamento de Física, Universidad de La Frontera, Casilla 54-D, Temuco, Chile
[2]Center for the Development of Nanoscience and Nanotechnology (CEDENNA), 9170124 Santiago, Chile
[3]Departamento de Física, Facultad de Ciencias, Universidad de Chile, Santiago, Chile
[4]Advanced Mining Technology Center (AMTC), Santiago, Chile
[5] UnB-Gama Campus, University of Brasilia, Brasilia DF 70910-900, Brazil.
[6] Plasma Physics Laboratory, Institute of Physics, University of Brasilia, Brasilia DF 70910-900, Brazil.
[7] University of Adelaide, School of Mathematical Sciences, Adelaide, SA 5005, Australia.
[8] National Institute for Space Research (INPE), Sao Jose dos Campos-SP 12227-010, Brazil

*Correspondence to* : eugenio.vogel@ufrontera.cl

**Abstract.** Four geographical zones are defined along the trench that is formed due to the subduction of the Nazca Plate underneath the South American plate; they are denoted A, B, C and D from North to South; zones A, B and D had a major earthquake after 2010 (Magnitude over 8.0), while zone C has not, thus offering a contrast for comparison. For each zone a sequence of intervals between consecutive seisms with magnitudes $\geq 3.0$ is set up and then characterized by Shannon entropy and mutability. These methods show correlation after a major earthquake in what is known as the aftershock regime, but show independence otherwise. Exponential adjustments for these parameters reveal that mutability offers a wider range for the parameters characterizing the recovery compared to the values of the parameters defining the background activity for each zone before a large earthquake. It is found that the background activity is particularly high for zone A, still recovering for zone B, reaching values similar to those of zone A in the case of zone C (without recent major earthquake) and oscillating around moderate values for zone D. It is discussed how this can be an indication for more risk of an important future seism in the cases of zones A and C. The similarities and differences between Shannon entropy and mutability are discussed and explained.

## I. INTRODUCTION

A recent advance on information theory techniques, with the introduction of the concept of mutability (Vogel et al., 2017a), opens new ways of looking at the tectonic dynamics in subduction zones. The main goals of the present paper are five-fold: 1) To establish the similarities and differences between mutability and the well-known Shannon entropy to deal with seismic data distributions; 2) To find out which of the aforementioned parameters gives an advantageous description of the subduction dynamics in order to discern different behaviors along the subduction trench; 3) To apply this description to characterize the recovery regime after a major earthquake; 4) To use this approach to establish background activity levels prior to major earthquakes; 5) To apply all of the above to different geographical zones looking for possible indications for regions with indicators pointing for possible future major earthquakes.

Several statistical and numeric techniques have been proposed to analyze seismic events. For a recent review we refer the interested reader to the paper by de Arcangelis et al. and references therein (de Arcangelis et al., 2016). We shall concentrate here in the use of Shannon entropy and mutability which are introduced and discussed in the next paragraphs; they will be applied to the intervals between consecutive seisms in each region.

Data may come from a variety of techniques used to record variations in some earth parameters like infrared spectrum recorded by satellites (Zhang et al., 2019), earth surface displacements measured by Global Positioning System (GPS) (Klein et al., 2018), variations of the earth magnetic field (Cordaro et al., 2018; Venegas-Aravena et al., 2019), changes in the Seismic Electric Signals (Varotsos et al., 1984a) 1984b); Varotsos et al., 1986; Varotsos et al., 1991; Varotsos et al., 1993; Varotsos et al. 2001; Varotsos et al., 2005; Sarlis et al., 2018; Varotsos et al. 2011c), Varotsos et al., 2019), among others. In the present work we make use of the seismic sequence itself as in natural time analysis (see, e.g., Varotsos et al. 2001; Varotsos et al. 2002; Varotsos et al. 2011a), 2011b)) analyzing the time intervals between filtered consecutive seisms.

Shannon entropy is a useful quantifier for assessing the information content of a complex system (Shannon, 1948). It

has been applied to study a variety of nonlinear dynamical phenomena such as magnetic systems, the rayleigh-Bernard convection, 3D MHD model of plasmas, turbulence or seismic time series, among others (Crisanti et al., 1994; Xi et al., 1995; Cakmur et al., 1997; Chian et al., 2010; Miranda et al., 2015; Manshour et al. 2009).

Analysis of the statistical mechanics of earthquakes can provide a physical rationale to the complex properties of seismic data frequently observed (Vallianatos et al., 2016). A number of studies have shown that the complexity in the content information of earthquakes can be elucidated by Shannon entropy. Telesca et al. (2004) applied Shannon entropy to study the 1983-2003 seismicity of Central Italy by comparing the full and the aftershock-depleted catalogues, and found a clear anomalous behaviour in stronger events, which is more evident in the full catalogue than in the aftershock-depleted one. De Santis et al. (2011) used Shannon entropy to interpret the physical meaning of the parameter b of the Gutenberg-Richter law that provides a cumulative frequency-magnitude relation for the statistics of the earthquake occurrence. Telesca et al. (2012) studied the interevent-time and interevent-distance series of seismic events in Egypt from 2004 to 2010, by varying the depth and the magnitude thresholds.

Telesca et al. (2013) combined the measures of the Shannon entropy power and the Fisher information measure to distinguish tsunamigenic and non-tsunamigenic earthquakes in a sample of major earthquakes. Telesca et al. (2014) applied the Fisher-Shannon method to confirm the correlation between the properties of the geoelectrical signals and crust deformation in three sites in Taiwan. Nicolis et al. (2015) adopted a combined Shannon entropy and wavelet-based approach to measure the spatial heterogeneity and complexity of spatial point patterns for a catalogue of earthquake events in Chile. Bressan et al. (2017) used Shannon entropy and fractal dimension to analyze seismic time series before and after eight moderate earthquakes in Northern Italy and Western Slovenia.

In the last two decades the concept of "natural time" for the study of earthquakes has been introduced by Varotsos et al. (Varotsos et al., 1984; Varotsos et al. 1991; Varotos et al. 1993; Varotsos et al., 2011) This method proposes a scaling of the time in a time series, by using the index $\chi_k = k/N$, where $k$ indicates the occurrence of the $k$-th event and $N$ is the total number of the events in a time series. For example, for seismic time series the evolution of the pair $(\chi_k, M_{0k})$ is following, where $M_{0k}$ is proportional to the energy released in an earthquake, finding interesting results in the Seismic Electric Signal previous to an earthquake occurrence (Sarlis et al., 2013; Sarlis et al., 2015; Sarlis et al., 2018a; Sarlis et al. 2018b; Rundle et al., 2018). An entropy has been defined in natural time (Varotsos et al., 2011b) -being dynamic and not static (Varotsos et al. 2003; Varotsos et al. 2007)- by $S \equiv \langle \chi ln(\chi) \rangle - \langle \chi \rangle ln \langle \chi \rangle$, and has been very useful in the analysis of global seismicity (Rundle et al., 2019).

On the other hand, the method based on information theory (Luenberg, 2006; Cover et al., 2006; Roederer 2005) was introduced a decade ago when it was successfully used to detect phase transitions in magnetism (Vogel et al., 2009; Vogel el al., 2012; Cortez et al., 2014). Then a new data compressor was designed to recognize compatible data, namely, data based on specific properties of the system. This method required comparing strings of fixed length and starting always at the same position within the digits defining the stored record. For this reason it was named "word length zipper" (wlzip for short) (Vogel et al., 2012). The successful application of wlzip to the 3D Edwards-Anderson model came immediately afterwards, where one highlight was the confirmation of a reentrant transition that is elusive for some of the other methods (Cortez et al., 2014). Another successful application to critical phenomena was for the disorder to nematic transition that occurs for the depositions of rods of length $k$ (in lattice units) on square lattices: for $k \geq 7$ one specific direction for depositions dominates over when deposition concentration overcomes a critical minimum value (Vogel et al., 2017b).

But wlzip proved to be useful not only for the case of phase transitions. It has been used in less drastic data evolution revealing different regimes or behaviors for a variety of systems. The first of such applications were in econophysics dealing with stock markets (Vogel et al., 2014) and pension systems (Vogel et al., 2015). The alteration of the blood pressure parameters was also investigated using wlzip (Contreras et al., 2016). At a completely different time scale the time series involved in wind energy production in Germany was investigated by wlzip yielded recognition of favorable periods for wind energy (Vogel et al., 2018).

The first application of wlzip to seismology came recently using data from a Chilean catalogue finding that wlzip results clearly increase several months prior to large earthquakes (Vogel et al., 2017a), thus being in accordance with natural time analysis which reveals (Varotsos et al., 2011b) that before major earthquakes there is a crucial time scale of around a few to several months in which changes in the correlation properties of physical quantities like seismicity or crustal deformation are observed. This early application of wlzip intended to establish the method without attempting further analyses or comparison with other methods or to compare possible seismic risk among regions, which are among the aims of the present paper.

In the present paper we make a new analysis comparing results from mutability and Shannon entropy applied to data of seismic data along the subduction front parallel to the Chilean coast. The complete tectonic context shows an active and complex seismic region for all the coast, driven by the convergence of the Nazca plate and the South American plate, at a rate of 68 mm yr$^{-1}$ (Altamimi et al., 2007) approximately. In the last 100 years, many large earthquakes have been localized in the shock between these two plates, such as Valparaíso 1906 ($M_w$=8.2), Valdivia 1960 ($M_w$=9.6), Cobquecura 2010 ($M_w = 8.8$), Iquique 2014 ($M_w$=8.2), and Illapel 2015 ($M_w$ =8.4). So, this zone

is an attractive source for studying seismic activity associated to large earthquakes. But although the dynamics along the Chilean coast may be dominated by the interaction between these two plates, various works have pointed out variations along the coast which may yield information about the details of that interaction. For instance, the coupling between these two plates has been studied by Metois et al. (Metois et al., 2012; Metois et al. 2013) in the last years, concluding that the subduction area has alternated zones of high an low coupling (Metois et al., 2012; Metois et al., 2013). This suggests that it is interesting to apply novel nonlinear techniques to study such variability. Here, we propose new ways to characterize some of the various dynamics that may be present along the subduction zone in this trench. In order to do that, we will consider four regions along the coast of Chile characterizing them mainly by their latitudes.

The paper is organized in the following way. Next Section is about methodology dealing with the data and parameters to be measured. Section 3 presents the results discussing them and comparing the alternative methods. Section 4 is devoted to conclusions.

## II. METHODOLOGY

### A. Data organization

Earthquakes originated in the subduction zone of the Nazca plate underneath the South American plate have been recorded, interpreted and stored in several data seismic data banks. In the present study we shall use the data collected by the Chilean National Seismic Centre (Centro Sismológico Nacional: CNS) (Web, 2019), which are very accurate regarding the location of the epicenters. In particular, we have used a seismic data set collected from March 2005 until March 2017, containing 22 697 events, distributed along the coast of Chile, from Arica in the far north up to Temuco in the south of Chile. These data are freely available through CNS (www.sismologia.cl).

In order to analyze the spatial evolution of the mutability and Shannon entropy along this part of the subduction zone, we have focused our attention on four regions defined below. For each region we have corroborated that the Gutenberg-Richter law holds, finding a common completeness magnitude of $M_w = 3.0$. Thus, all the following analysis will be made using only the seismic events with magnitudes of at least $M_w = 3.0$. We have considered seismic data sequences for four specific geographical zones: three of them include one earthquake over 8.0 occurring after 2010, and we have added for comparison a neighboring area with no such large earthquake during several recent years.

Starting from the North, the zones are the following: A) around the earthquake near Iquique (2014; $M_w = 8.2$) including 6891 events; B) around the earthquake near Illapel (2015; $M_w = 8.4$) including 6626 events, C) a quieter geographical region (calm zone) at the center of Chile (where the greatest seismic event is $M_w = 6.5$), including 2824 events; and D) around the earthquake in Cobquecura (2010, $M_w = 8.8$) including 6356 events. The observation time is from January 1, 2011 to March 23, 2017 for zones A, B and C, while it is from January 1, 2009 to March 23, 2017 for zone D (no special reason for this last date). We extended the analysis in the case of zone D to include the regime previous to the big earthquake of 2010. Since the analysis is either relative to the size of the sample or dynamical along the series this difference should not affect the discussion below.

All zones have a similar geographical extension with some singularities that we explain here. Regions A, B, and D have latitudes centered at the epicenter of the largest earthquake of each zone; the span in longitude is the same for these zones. Zone A misses the 4.0° spans in latitude of zones B and D, since the Chilean catalogue ends at $-17.926°$ which is the northern limit for this zone (for homogeneity of the data we do not mix catalogues). The largest span for the zones under study is 4 degrees in each direction; it was chosen as a mean to consider enough data within each zone in order to have good statistics. On the other hand, zone C was chosen to include a populated area of the country but with no earthquake over 8.0 and showing less important activity than previous ones. Details are given in Table I, and are illustrated in Fig. 1. As it can be seen in this map zone C overlaps with both B and D: to avoid getting close to the epicenter of the main earthquake in zone D, zone C was shortened in its South extension. So the data catalogues have been filtered by latitude, longitude and magnitude. At this point we do not filter by depth which should not greatly influence the comparison among zones since it is a common criteria for all of them.

Originally the study contemplated zones A, B and D only concentrating on the main three earthquakes of the decade. In spite the main purposes of this work are accomplished by looking at zones A, B, and D, only, we decided to broaden the geographical coverage a bit. The area in between zones B and D could be unstable as subductions took place both south and north of it. Eventually, the subduction here is stuck and it could be interesting to find out the behavior of this densely populated zone. We paid the price of overlapping with the neighboring zones but special care has been taken as to avoid in zone C the epicenters and immediate vicinity of the major earthquakes and to initiate the analysis in 2011, several months after the largest earthquake included in this study.

For all seismic events characterized above we calculate the interval in minutes (rounding off seconds) between consecutive events. Then a vector file is produced storing the consecutive intervals between theses seisms within each

| Zone | Latitudes | | Longitudes | | Main Earthquake | | | |
|------|-----------|------|------------|------|------|------|---|---|
| | N | S | W | E | Magn | Y | M | D |
| A | $-17.926°$ | $-21.572°$ | $-75.00°$ | $-68.00°$ | 8.2 | 2014 | 4 | 1 |
| B | $-29.637°$ | $-33.637°$ | $-75.00°$ | $-68.00°$ | 8.4 | 2015 | 9 | 16 |
| C | $-32.700°$ | $-35.500°$ | $-74.00°$ | $-69.00°$ | (6.5) | 2012 | 4 | 17 |
| D | $-34.290°$ | $-38.290°$ | $-75.00°$ | $-68.00°$ | 8.8 | 2010 | 2 | 27 |

TABLE I. Geographical definition of the 4 zones considered in this study. The strongest seismic event in each zone is identified at the end. Zone C lacks a very strong seism during recent years which is indicated by the use of parenthesis for the strongest seism here. The geographical coordinates and time windows are explained and defined in the text.

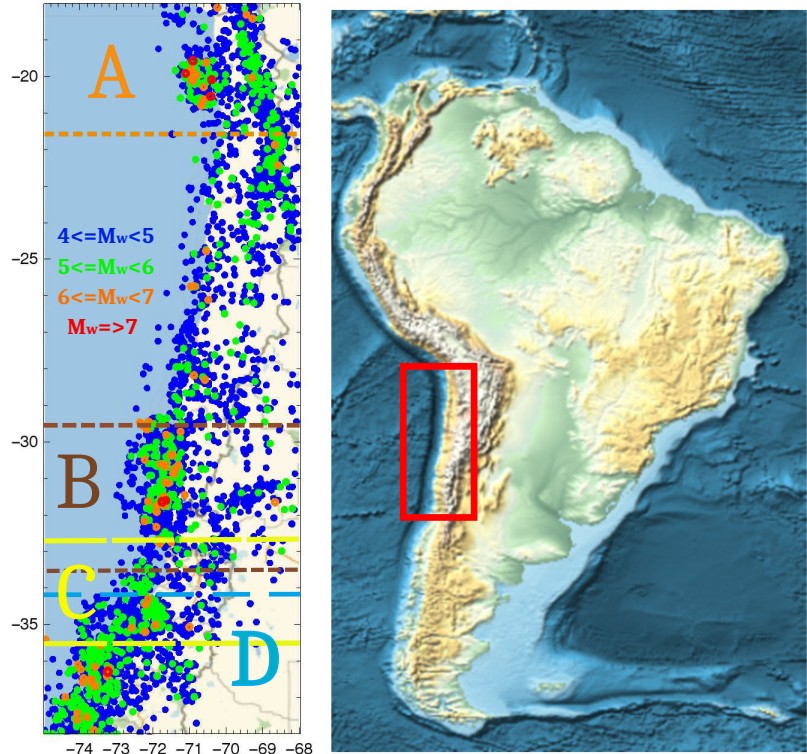

FIG. 1. Left: Map showing the seismic events with magnitudes greater than $M_w$ 4.0 and the division in four geographical zones A, B, C, and D defined in Table I. The seismic events are shown by circles using the following color code according to magnitude: between 4.0 and 4.9 in blue, between 5.0 and 5.9 in green, between 6.0 and 6.9 in orange, and for magnitude equal or greater than $M_w$ 7.0 in red. Right: Map of South America showing by a red rectangle the area displayed in the figure to the left. The trench between the Southamerican plate and the Nazca plate appears in dark blue.

zone. These are the files to be analyzed by Shannon entropy and mutability. Notice that there is a close similarity between this and the "natural time" analysis discussed in the Introduction, since the resulting vector is indexed by the event number. In our case, the value of each vector component is the interevent time itself, which has been also used in the natural time analysis of electrocardiograms by considering the interevent time between consecutive heartbeats (Varotsos et al., 2007). Registers in the vector storing the information in our analysis are the interevent intervals, thus temporal information is still kept in the time series.

Let us consider histograms for interval distributions for each zone with consecutive bins of 60 minutes each. Percentage of abundance $G_{K,i}$ of intervals are obtained for the $i$-th bin for the different zones $Z$: A, B, C, or D. Figure 2 shows the histograms corresponding to the distribution functions $G_{Z,i}$. It can be immediately seen that shorter intervals have been more frequent in zones D and B, while they are less frequent in the $C$ zone. Zone A presents and intermediate presence of small intervals. This different frequency for small seisms finds an explanation in the presence of large earthquakes in $B$ and $D$ followed by large aftershock periods, while in zone $A$ the aftershock period (and the number of short intervals) was very short as we will see in detail below; zone $C$ does not include any aftershock period so short intervals are less frequent here.

To better establish the role of the aftershocks we compared the number of seisms (3.0+) on the month prior to the

largest earthquake in the zone and the number of seisms in the same zone during the month after it. For zone B these numbers are 49 and 1439 respectively; for zone D these numbers are 11 and 1006 respectively. This comparison with the background assures the large production of aftershock seisms. This comparison is not possible for zone A since the main earthquake came during the aftershock period of a large precursor as discussed below. In addition we did a restricted geographical analysis for the month after each main earthquake comparing the number of seisms in the full zone to the number of seisms in a smaller zone of two degrees in each direction around the epicenter of the main seism. For zone A we have 939 and 736; for zone B 1439 and 1147; for zone D 1006 and 787. It is clear that the largest amount of seisms occurred in the vicinity and in the days after the largest earthquakes in each zone.

These plots are presented in a semilog scale to better appreciate any possible decay law. However, no general behavior is found evidencing the different dynamics among the zones. Zone A presents a linear decay in this scale while zone C is the more irregular one. On the other hand, zone D departs quite clearly from a linear dependence evidencing the lack of saturation several years after the huge earthquake of 2010. Scaling algorithms have been suggested to deal with the time series on the interevent sequence (Lippiello et al. 2012) but in the present study we leave the series with the natural interevent intervals to analyze them by means of information theory as proposed below.

We can increase the precision of the data treatment below by the use of a database providing more positions for the numeric recognition (Vogel el al., 2017a). This was achieved by choosing a numerical basis providing more positions to be matched. So the data files used both for Shannon entropy and for mutability used digits corresponding to a quaternary numerical basis.

## B.  Shannon entropy

Let $\Delta_i$, $i = 1, ..., N$ be the full sequence of time intervals between consecutive seisms in any of the already defined zones. The time that the $i$-th event occurred can be obtained by $t_i = t_0 + \sum_{j=1}^{i} \Delta_j$, where $t_0$ is the start time of the dataset. The Shannon entropy for $\Delta_i$ within a sliding window of size $\nu$ events can be calculated as follows

$$H(t_i, \nu) = -\sum_{j=i}^{i+\nu} p_j \ln(p_j) \tag{1}$$

where $p_j$ is the probability distribution function of the time intervals within the time window, which can be determined by constructing a normalized histogram

$$p_j = g_j/\nu \tag{2}$$

where $g_j$ is the number of times $\Delta_j$ occurs within the sliding window. The appropriate value for $\nu$ depends on the kind of data under consideration. Thus, for instance, application to this method to the minute variations of the stock market yielded $\nu = 30$ (half-an-hour) as a significant time window to establish tendencies in this economical activity (Vogel et. al 2014). In the case of seismic sequences ordered by real time, time windows between $\nu = 24$ and $\nu = 96$ were investigated finding that $\nu = 24$ is appropriate to deal with seismic activity (Vogel et. al 2017a). More details about the choice of $\nu$ can be found in these references in particular in Fig. 3 of the last reference. So, for all applications below we use $\nu = 24$.

## C.  Data recognizer

We use here the same dynamical data window of $\nu$ events used for the calculation of Shannon entropy. The weight in bytes of the sequence of $\nu$ events beginning at natural time $i$ will be denoted by $w(t_i, \nu)$. This partial sequence is processed by wlzip producing a new sequence that needs $w^*(t_i, \nu)$ bytes of storage. The relative dynamic information content of this time series of seismic events is known as mutability, which is defined as

$$\mu(t_i, \nu) = \frac{w^*(t_i, \nu)}{w(t_i, \nu)}, \tag{3}$$

where $w^*$ is the size in bytes of the compressed dataset associated to the time intervals $\Delta_j$ within the time window of $\nu$ events.

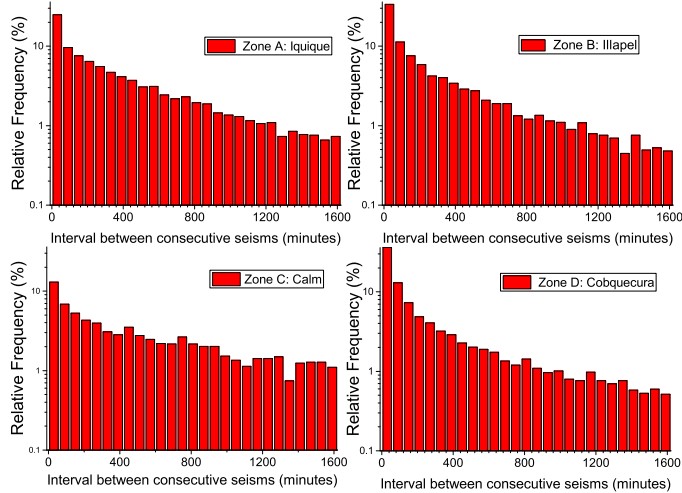

FIG. 2. colorcyan Distribution functions $G_{Z,i}$ ($Z = \{$A, B, C, D $\}$) for intervals between two consecutive seismic events.

As already pointed out $\nu = 24$ for all mutability calculations below. The typical value of $w(t_i, \nu)$ for the files measured here is 144 bytes, while the values of $w^*(t_i, \nu)$ vary roughly between 100 to 400 bytes thus leading to variations in mutability. Mutability is a relative measure of the information content present in a file: monotonic sequences give low mutability values; chaotic sequences (like those emanating from phase transitions) give high mutability values. Its dynamic response is advantageous to detect information content even when other methods fail; an example of this is the Edwards-Anderson model where spin-glass transitions are revealed by mutability in spite magnetization measurements fail (Cortez et al. 2014). To better illustrate this concept we include an Appendix calculating the mutability for 4 different sequences of 24 events.

Two comments are in order: First, wlzip uses compressor algorithms to recognize information but this does not mean that $w^*(t_i, \nu)$ should be less than $w(t_i, \nu)$; Second, the value of wlzip depends both on the interval distribution but also on the time sequence of the intervals, which has been also used in the natural time analysis of the consecutive heartbeat intervals, while Shannon entropy depends only on the distribution (Varotsos et al., 2007). Thus, the sooner a value in the sequence is repeated, the lower the value of $\mu(t_i, \nu)$ is (Vogel et al., 2012; Cortez et al., 2014). This fact marks a difference between these two parameters as we will see below.

## III. RESULTS

Figs. 3, 4, 5 and 6 present the Shannon entropy (top) and mutability (bottom) for data corresponding to geographical areas A, B, C and D respectively according to Table I and Fig. 1. The numeric recognition was done for the data files (intervals in minutes between successive seisms) in quaternary basis both for Shannon entropy and mutability. All registers have the same number of digits filling with zeroes all empty positions previous to first significant digit, The matching to recognize the same data register started at position 4 and was done for three digits including the fourth position (Vogel et al., 2017a). All zones were treated with the same precision.

In the upper panel the abscissa "Time" corresponds to real time $t_i$ (as defined in Section II. B) beginning on January 1, 2011 for zones A, B and C, or on January 1, 2009 for zone D. In the lower panel the abscissa labelled "Events" corresponds now to the succession of filtered seisms identified by the same label $i$ used to define $t_i$. The ordinates are the same in both panels.

In the upper panel the aftershock behavior is concealed by the large activity in the short time after a large quake, while in the lower panel it is easier to see the aftershock sequence although the large quiet periods look now more compressed. Earthquakes over a certain magnitude (as given in the inset for each zone) are marked by a star. The empty square (A, B, and D only) identifies the largest earthquake with magnitude greater than $M_w = 8.0$ within that area as listed in Table I.

To facilitate the interpretation of these figures and the interrelation between both abscissa axes in each figure, Table II interprets the actual real time for the milestones 1000, 2000, ....6000 events, for the four zones. Date is given by

|  | Zone A | | | Zone B | | | Zone C | | | Zone D | | |
|---|---|---|---|---|---|---|---|---|---|---|---|---|
| Event | Y | M | D | Y | M | D | Y | M | D | Y | M | D |
| 1000 | 2012 | 2 | 29 | 2012 | 10 | 16 | 2012 | 5 | 22 | 2010 | 3 | 22 |
| 2000 | 2013 | 6 | 2 | 2014 | 8 | 19 | 2014 | 10 | 22 | 2010 | 5 | 17 |
| 3000 | 2014 | 3 | 24 | 2015 | 9 | 19 | - | - | - | 2010 | 9 | 25 |
| 4000 | 2014 | 4 | 26 | 2015 | 10 | 14 | - | - | - | 2011 | 7 | 30 |
| 5000 | 2015 | 1 | 6 | 2016 | 1 | 28 | - | - | - | 2012 | 12 | 30 |
| 6000 | 2016 | 3 | 7 | 2016 | 9 | 1 | - | - | - | 2015 | 12 | 15 |

TABLE II. Equivalence of the milestones for natural time in thousand of events in terms of real date: year (Y), month (M) and day (D) for figures 3, 4, 5, and 6.

year (Y), month (M), and day (D).

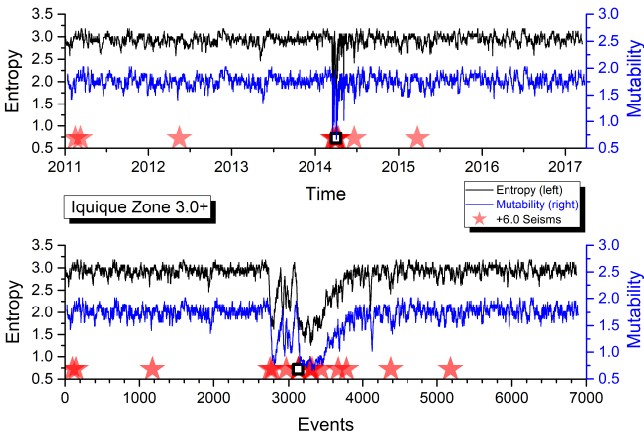

FIG. 3. Shannon entropy and mutability as functions of time for the seismic activity of the A zone. The open star marks the position of the earthquake identified in Table I. The abscissa in the upper panel corresponds to real time $t_i$ while in the lower panel it represents natural time or successive events (filtered seisms) denoted by the order label $i$ (Figs. 4, 5, and 6 use the same procedure).

As it can be observed, both $H$ and $\mu$ present a similar behavior for the data in the four areas. Immediately after a large shock both indicators sharply decrease due to the short intervals between consecutive aftershock quakes thereafter.

The average activity level is relatively constant before a major earthquake and later on after the aftershocks have disappeared. However, such activity level is not the same for all the areas which is an indication of different response to similar phenomena which deserves particular attention and it will be further investigated below.

To better appreciate the correlation between $H$ and $\mu$ we study the out-of-phase correlations defined as

$$C_H(\ell) = \frac{1}{(N - 2m - 1)\sigma_H \sigma_\mu} \sum_{i=m+1}^{i=N-m} [H(i) - \bar{H}][\mu(i - \ell) - \bar{\mu}] \tag{4}$$

$$C_\mu(\ell) = \frac{1}{(N - 2m - 1)\sigma_H \sigma_\mu} \sum_{i=m+1}^{i=N-m} [H(i - \ell) - \bar{H}][\mu(i) - \bar{\mu}] \tag{5}$$

where $\ell$ is the phase difference measured in terms of number of events separating the measurement of one parameter with respect to the other and $m = 50$ is the range or maximum phase difference in either sense considered here. This value is entirely empirical looking for a flat behavior of previously defined correlations. From Fig. 7 it may appear that $m = 30$ could be enough but we decided to explore a bit further to make sure curves are already tending to a flat behavior. Previous equations represent the average over the $(N - 2m - 1)$ possible equivalent ranges within the

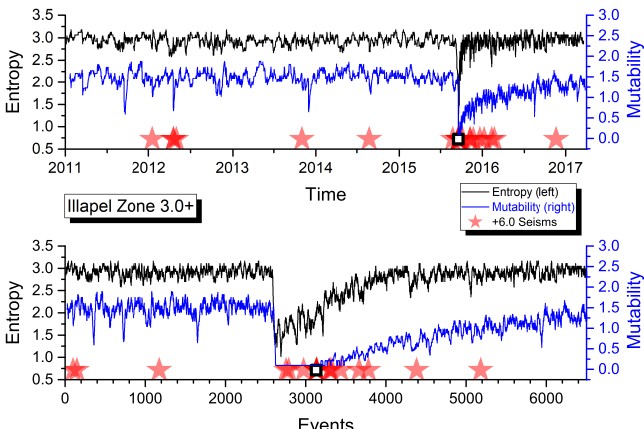

FIG. 4. Shannon entropy and mutability as functions of real time (top) and natural time (or sequence of events, bottom) for the seismic activity of the B zone. The open star marks the position of the earthquake identified in Table I.

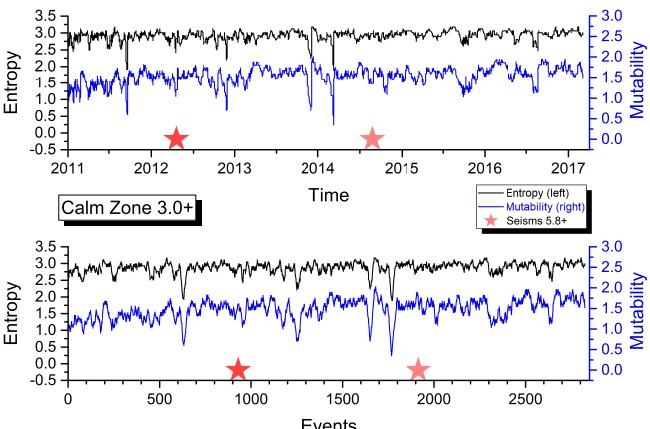

FIG. 5. Shannon entropy and mutability as functions of real time (top) and natural time (or sequence of events, bottom) for the seismic activity of the C zone.

series on $N$ registers. In addition $\sigma_H$ and $\sigma_\mu$ represent the standard deviations of $H$ and $\mu$ through the N events respectively.

The out-of-phase correlation between Shannon entropy and mutability is presented in Fig. 7: it was found that in general full correlation is lost after about 20 events. A general prevalence is observed in the form of a tendency towards a constant behavior far from the maximum: a value around 0.75 in the wings of zone B (top panel) and towards 0.15 for the zone C (middle panel). Similar figures were analyzed for zones A and D with prevalence values near 0.75 and 0.57 respectively. To test if these prevalence correlations are due to the aftershock regimes a reevaluation of the out-of-phase correlation was done for the D zone restricted to results of Shannon entropy and mutability obtained after January 1, 2013, thus diminishing the effect of the aftershock regime; these results are also shown in Fig. 7 (lower panel). So the main correlation between Shannon entropy and mutability is obtained during the aftershock period. On the other hand the out-of-phase correlation tend to be completely lost during periods without the influence of this regime. This is a first indication for partial independence between Shannon entropy and mutability.

The recovery of the activity level after a major earthquake is faster for the Shannon entropy than for the mutability. Namely, the slope in the recovery for $\mu$ is better defined after a large quake. It is interesting to notice from figures 3 through 6 that zone A recovered its foreshock activity level sooner than any of the other zones. This observation will be put in a quantitative way concentrating on the recovery dynamics in real time to compare the behavior of the different zones.

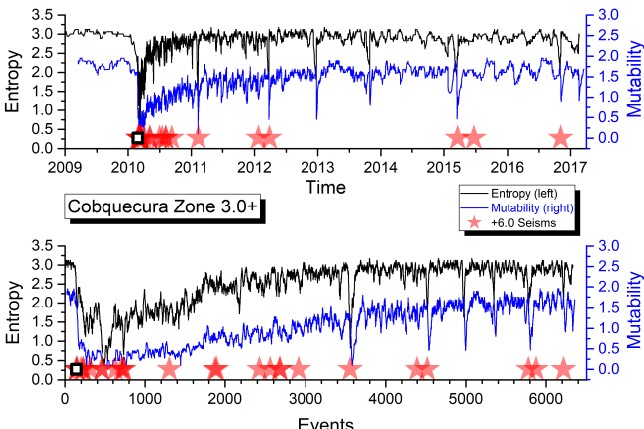

FIG. 6. Shannon entropy and mutability as functions of real time (top) and natural time (or sequence of events, bottom) for the seismic activity of the D zone. The open star marks the position of the earthquake identified in Table I.

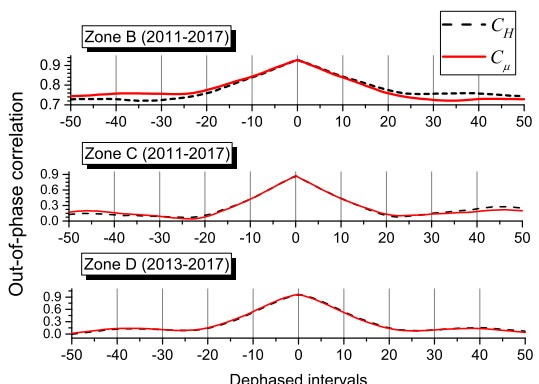

FIG. 7. Out-of-phase correlations. Upper panel: B zone data including aftershock regime (similar ones are obtained for zones A and D with aftershock regimes). Middle panel: C zone data that does not present aftershock regime. Lower panel: Truncated D zone data to exclude the aftershock regime.

Figure 8 presents the mutability results for region D starting at the point of minimum mutability occurring immediately after the major earthquake on February 27, 2010. The dotted (red) curve corresponds to an exponential fit to be discussed next. The inset shows the same data and exponential adjustment for the first two years on the time span. A power law can be seen at the onset of the aftershock regime resembling Omori's law.

For zones A, B, and D, we assume an exponential adjustment for the mutability function after the largest earthquake. A possible such function is:

$$\mu_{eZ}(t) = a_Z + b_Z \exp(-(t - t_Z)/\tau_Z), \tag{6}$$

where $a_Z$ measures the "asymptotic" activity of zone $Z$ (reached after the aftershocks regime), $t_Z$ corresponds to the time of minimum mutability after the largest earthquake (Table I) and serves as initial time for this recovery analysis; $\tau_Z$ is the characteristic time for activity recovery in zone $Z$. $b_Z$ is just a shape adjustment parameter without a direct meaning for this analysis.

For zone D (Fig. 8), the best least square fit for the mutability is obtained for $a_D = 1.502(2)$ and $\tau_D = 0.62212$ years (y). The results of this treatment for all the zones with major earthquakes are summarized in Table III. Fig. 8 includes an inset with semilog scale to appreciate the recovery process under a different perspective. A linear behavior in this scale is apparent at the beginning of the plot, but then it is rapidly lost. The sudden decrease of mutability values during February 2011 is better resolved in the time scale of the inset: this is due to the short aftershock activity produced by an earthquake of magnitude 6.1 occurred on February 14, 2011. Due to their sharp appearance we

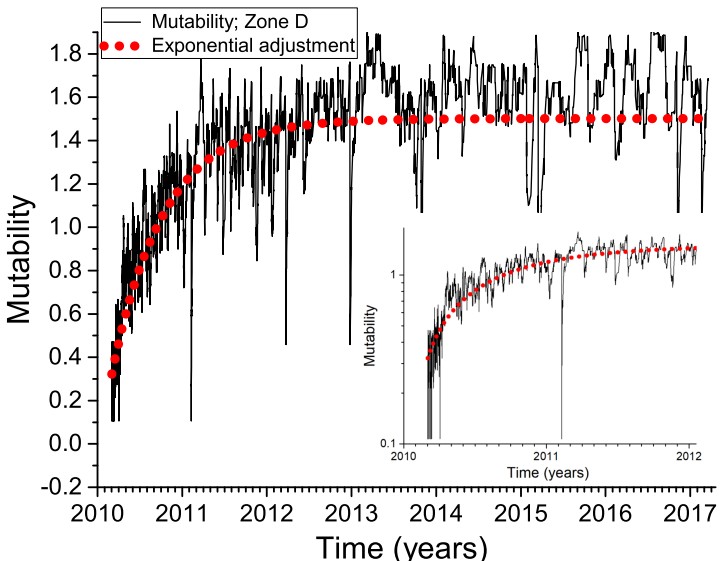

FIG. 8. Exponential fit for Cobquecura data set after February 27, 2010. This data set starts at the point of minimum mutability after the big earthquake of magnitude $M_w = 8.8$.

<sup>297</sup> propose to call "needles" these sudden and short decreases of mutability associated to the brief aftershock period left
<sup>298</sup> by seisms M5.0 to M6.0 approximately. Other needles can be easily spotted in Figs. 3, 4, 5, 6 and 8.

| Zone $Z$ | $a_Z$ | $b_Z$ | $t_Z$ (y) | $\tau_Z$ (y) |
|---|---|---|---|---|
| A | 1.754 (0.002) | $-1.64691$ | 2014.24829 | 0.0134(3) |
| B | 1.208 (0.004) | $-1.09124$ | 2015.70784 | 0.2092(33) |
| D | 1.502 (0.005) | $-1.37833$ | 2010.07093 | 0.6221(110) |

TABLE III. Best fit parameters for the mutability of zones A, B, and D, after the main earthquake, using the exponential trial function given by Eq. (6).

<sup>299</sup>    A similar analysis was made for the Shannon entropy results using the same exponential fit and the corresponding
<sup>300</sup> parameters are given in Table IV.

| Zone $Z$ | $a_Z$ | $b_Z$ | $t_Z$ (y) | $\tau_Z$ (y) |
|---|---|---|---|---|
| A | 2.924(2) | $-3.69218$ | 2014.24516 | 0.0095(3) |
| B | 2.908(3) | $-2.30957$ | 2015.69997 | 0.0246(4) |
| D | 2.815(4) | $-2.29226$ | 2010.13133 | 0.1255(25) |

TABLE IV. Best fit parameters for Shannon entropy of zones A, B, and D, after the main earthquake, using the exponential trial function Eq. (6).

<sup>301</sup>    Figures similar to Fig. 8 were made for the mutability of zones A and B using the best fit parameters listed in
<sup>302</sup> Table III. The same analysis was also done for the results obtained by Shannon entropy and the corresponding
<sup>303</sup> parameters are given in Table IV. The figures backing such fittings are not included here since they are very similar
<sup>304</sup> to Fig. 8 and the procedure is the same to the one already established in the presentation of this figure.
<sup>305</sup>    Let us now discuss the results given in Tables III and IV which list the parameters defined in Eq. 6. The first
<sup>306</sup> striking difference between Shannon entropy and mutability is on the value for the background parameter $a_Z$. In
<sup>307</sup> the case of the adjustment for Shannon entropy this parameter does not discriminate significantly among zones with
<sup>308</sup> values close to 2.9 for all of them; the same parameter in the case of the mutability data spans a range [1.208,1.754],
<sup>309</sup> thus indicating differences in the dynamics of these three regions. In particular, mutability indicates that in zone B
<sup>310</sup> there are more seismic events at regular intervals than in the other zones. Given the underlying plate subduction
<sup>311</sup> mechanism, this could mean that plates are sliding more regularly or even fluently in zone B, whereas the relative

motion of the Nazca plate under the South-American plate is more difficult in zone A, thus leading to more disperse
set of intervals between consecutive seisms.

After a large earthquake the zones tend to recover their characteristic activity level $a_Z$, but this is done rather
abruptly for Shannon entropy while it is more gradual for mutability. This is measured by the recovery time $\tau_Z$ in
Tables III and IV. In the case of the Shannon entropy for the zone A the recovery is very fast, namely 0.00947 years
$\approx$ 3.5 days. In the case of zones B and D the recovery times for the Shannon entropy are of 9 days and 45 days,
respectively. However, when the analysis is done using the recovery time for mutability (Table III) the recovery times
are 5 days, 2.5 months and 7.5 months for the zones A, B, and D, respectively.

Tables III and IV also show that recovery times $\tau_Z$ are different, shorter for Shannon entropy and longer for
mutability, but the tendencies are the same. So eventually both methods can be used to characterize this aspect of
the aftershock regime. In terms of the human perception experienced after any large earthquake it seems that $\tau_Z$
values obtained for the mutability results are more representative of the aftershock times experienced in each zone.
Thus, for instance, seisms of magnitude around 4.0 were frequent in zone D during several months after February 10,
2010, but this was not the case for zone A where people lost perception of the aftershock regime after a week or so of
the last earthquake in this area.

The main difference between Shannon entropy and mutability is that the former analyzes the distribution of registers
in a sequence regardless of the order in which these entries were obtained, while the latter gives a lower result for
sequences including frequently repeated registers (Cortez et al., 2014). Shannon entropy considers the visit to a state
without considering the order in which these visits take place, which is of paramount importance for the entropy
in natural time being dynamic entropy and not a static one (Varotsos et al., 2005; Varotsos et al., 2007) so it pays
exclusive attention to the probability of visiting a state at some instance during the observation time. Mutability
considers also the trajectory in which these visits take place, giving lower results when the system stays long periods
in the same state or states directly connected to this state; on the contrary during agitated periods (chaotic dynamics
would the at the apex here) mutability gives higher results. In other words, a given sequence has just one result for
Shannon entropy but the permutations of the order of the registers lead to different results for mutability; in the
present case the mutability results reported here corresponds to the natural sequence of the recorded seisms.

We now focus on the analysis of the background activity obtained for the 4 zones described in this work, taking
semestral averages of the values of mutability in Figs. 3–6, in order to study trends in time scales longer than the
one of previous figures. We have chosen a semester as the time for averages so we have a few hundreds registers in
each partial sequence minimizing error, but still we have some 13 points in the overall period to appreciate tendencies
and differences. In doing so, we also evaluate semestral averages of intervals between consecutive seisms, which show
similar trends to the mutability results for the same period.

The semestral analysis for zones A, B, C and D is shown in Figs. 9, 10, 11 and 12, respectively; they are all
presented under the same scale to allow a direct comparison. The mutability values run on the upper part (black)
while the intervals tend to occupy the lower part (blue) of the plot. The first comment here is evident: these 4
regions present different responses to the evaluation of their sequences of time intervals between consecutive seisms
of magnitude 3.0+ as measured both by mutability and Shannon entropy. These two measures are not equivalent
either although some general similarity between them can be noticed. The only effective common feature is that an
earthquake with magnitude over 8.0 produces an absolute minimum for each variables during the semester containing
this seism and its aftershock sequence.

For didactical reasons we shall perform this discussion beginning with zone D, where the long recovery period
already appreciated in Fig. 6 and in Table III is more enhanced. It is interesting to observe that the average semestral
mutability presents recent relaxations like in the first semester of 2015 and the first semester of 2017. Generally
speaking these results do not approach yet the values near 1.8 for the average semestral mutability in the foreshock
period preceding the large earthquake of 2010. Interval semestral averages tend to follow the variations of mutability
but some differences are noticed. The present average interval of about 2000 minutes (about 33 hours) is far from the
almost 6000 minute interval before the large earthquake.

Fig. 11 is completely different to the others. There is no major earthquake included here but it is obvious that there
was one prior to 2011 from which this activity is slowly recovering. The general tendency is to slowly increase the
mutability values to levels similar to those constantly presented by zone A and those presented by region D previous
to the large earthquake. Interval averages also increase reaching just under 2000 minutes. If this is an announcement
for a future major earthquake in zone C or nearby is still too early to tell but this zone should be monitored closely.

Fig. 10 shows the foreshock mutability averages for zone B which present a nearly flat behavior around 1.6 before
the major earthquake of 2015. Then, after the aftershock regime the average semestral mutability begins to recover,
faster than in zone D, but still not reaching the level shown here previous to the large earthquake. The observation
is similar for the interval semestral average whose value is still small compared to the activity before 2016.

Fig. 9 shows the almost constant results (near 1.8) for the average semestral mutability of zone A, with just one
semester reaching a moderate low value (1.4 with large error bar). The semestral average for intervals between seisms

is also rather flat around 10 hours. The only exception is the first semester of 2014 in coincidence with the large earthquake there.

Error bars deserve a separate discussion. They are obtained from the standard deviations calculated for the distributions of each semester within each zone. So the number of events differ from one semester to another even within each zone. In the case of intervals the largest semestral error is of 4966 minutes for zone D during the second semester of 2009, just prior to the large earthquake of 2010. The smallest error is of 280 minutes obtained for the first semester of 2014, which includes the large earthquake and related activity in zone A. In the case of mutability its largest semestral error is for zone A during the first semester of 2014, while the smallest one is during the second semester of 2013 for this same zone. So error bars are subject to some fluctuations also but still they are a general indication for the homogeneity of the data.

Mutability error bars are rather small for the A zone, meaning that the intervals are rather similar along the data sequence. This is reinforced by the average interval error bars which are the smallest among the four zones (spanning only about 1200 minutes) telling that intervals are not so different among themselves. The largest error bars both for mutability and intervals are to be found in zone D; moreover they are irregular in recent years. Error bars increased for the average in zone D during 2009 just prior to the huge quake of 2010. However, for this same zone the corresponding error bars for the average semestral mutability are among the smallest to be found prior to this large earthquake. Once again it is difficult to say something about the present status of zone B since it is clearly under recovery. However, the Calm zone C is clearly showing a tendency: error bars for mutability averages are shrinking, while error bars for intervals are growing spanning about 60 hours. These two symptoms were present in zones A, B and D previous to their large respective earthquakes. In the case of zone A the error bars for the average intervals are not so large, but here is where we find the highest values for mutability and the smallest error bars for this variable.

If we look for common features just before a large earthquake they are: relatively high mutability values ("high" needs to be defined for each zone) and very small error bars associated with semestral mutability averages. The particular values of these indicators for zone A could be interpreted as an irregular subduction here, with no short-time accommodations or lack of fluency, leading to seismic risk of some sort, although it is not possible to specify any possible time for a large seism in the future. From this point of view, the earthquake of 2014 near Iquique was just a small accommodation of the plates but the subduction process could be somewhat stuck to the similar levels presented before the large quake.

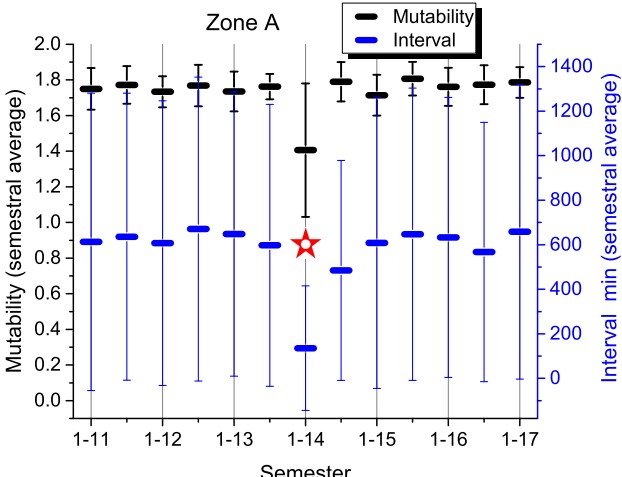

FIG. 9. Semestral average of mutability values (upper symbols; black) and intervals in minutes between consecutive seisms (lower symbols; blue) for zone A: Iquique. Odd semesters are labeled on the abscissa axis (1-13: first semester of 2013) while even semesters are only marked. A star identifies semester with earthquake of magnitude over 8.0.

## IV. CONCLUSIONS

Seismic activity is different for the four zones defined here along the Nazca-South American subduction trench (Figs. 1-2, Table I). Nevertheless, some general behaviors are common to the seismicity of the tectonic activity present in this region. Both Shannon entropy and mutability show a sudden decrease after an earthquake of magnitude around

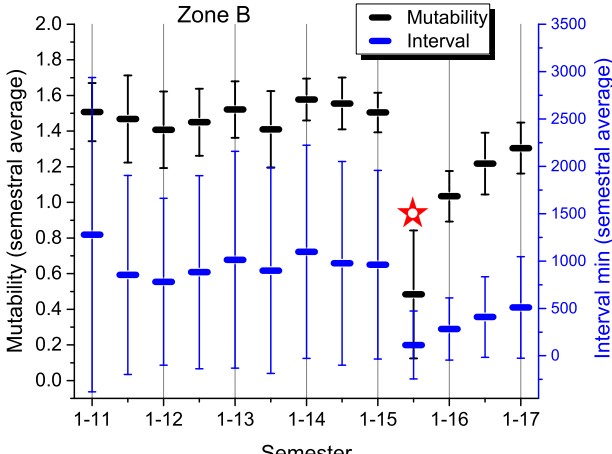

FIG. 10. Semestral average of mutability values (upper symbols; black) and intervals in minutes between consecutive seisms (lower symbols; blue) for zone B: Illapel. Odd semesters are labeled on the abscissa axis (1-13: first semester of 2013) while even semesters are only marked. A star identifies semester with earthquake of magnitude over 8.0.

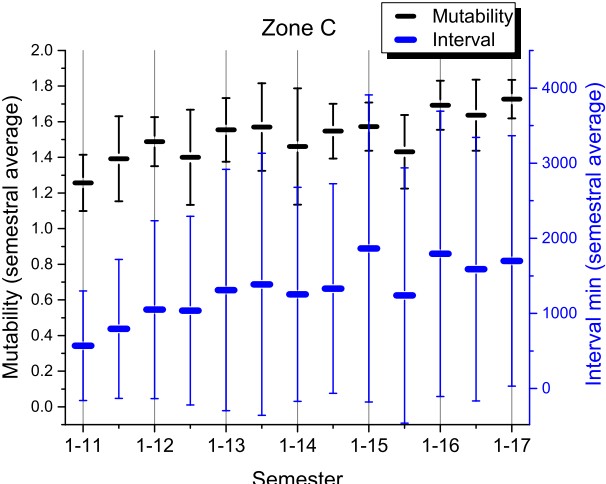

FIG. 11. Semestral average of mutability values (upper symbols; black) and intervals in minutes between consecutive seisms (lower symbols; blue) for zone C: Calm. Odd semesters are labeled on the abscissa axis (1-13: first semester of 2013) while even semesters are only marked.

₄₀₂ or over 7.0 (Figs. 3-6). Additionally, Shannon entropy and mutability reach "high" values before a major earthquake; ₄₀₃ the scale to define "high" needs to be tuned for each geographical region and observation time window.

₄₀₄ A short time correlation exists between Shannon entropy and mutability during the aftershock regime. However, ₄₀₅ this correlation is lost far from this regime thus providing independent tests to characterize the seismic activity (Fig. ₄₀₆ 7).

₄₀₇ The aftershock regime is characterized by successions of low and medium intensity seisms at short intervals producing ₄₀₈ low values of both Shannon entropy and mutability. After some recovery time the intervals tend to go back to the kind ₄₀₉ of intervals present before the large quake. This recovery behavior can be described by exponential adjustments (Fig. ₄₁₀ 8) which indicate that the characteristic times are longer for mutability than for Shannon entropy (Tables III-IV); ₄₁₁ eventually this speaks in favor of the former to continue the analysis. Another advantage of mutability is that the ₄₁₂ parameter reflecting the background activity span larger ranges than the one presented by the adjustment of Shannon ₄₁₃ entropy (Tables III-IV). From these results the mutability recovery time $\tau_Z$ for zone A lasted a few days, while the ₄₁₄ same parameters for zone D lasted several months, which is close to the human perception in these zones.

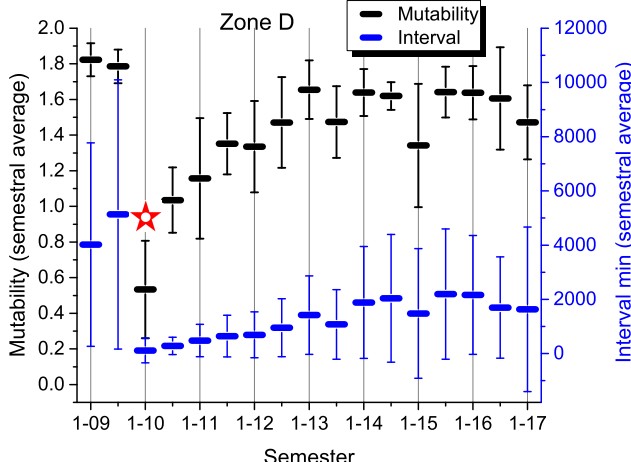

FIG. 12. Semestral average of mutability values (upper symbols; black) and intervals in minutes between consecutive seisms (lower symbols; blue) for zone D: Cobquecura. Odd semesters are labeled on the abscissa axis (1-13: first semester of 2013) while even semesters are only marked. A star identifies semester with earthquake of magnitude over 8.0.

The difference between Shannon entropy and mutability evidenced after the recovery time are due to the handling of a static distribution by the former while the latter considers the order in which registers entered in the distribution in accordance with the concept of natural time. The differences between Shannon entropy and mutability evidenced after the recovery time are due to the handling of a static distribution by the former while the latter considers exact or approximate repetitions in the data chain. From this point of view mutability carries more information than Shannon entropy in spite both are obtained from the same sequences.

The background activity based on mutability $a_Z$ (Tables III-IV) is quite different for each zone (Figs. 9-12). This means that the subduction process finds different difficulties in each zone. However, some general features describing the motion of the Nazca plate under the South-American plate should be present along the trench. To investigate this possibility we considered semestral averages of mutability values.

Semestral averages for mutability recovered soon for zone A after the 8.2 earthquake, which indicates that the short intervals after a major earthquake were mostly absent here. Soon, the regime with longer and different intervals reappeared raising the values of mutability and narrowing the corresponding error bars; this could be interpreted as a warning for a possible earthquake in this zone sometime in the near future. On the opposite side is zone D where the semestral averages still do not recover to the levels prior to the large 8.8 earthquake of 2010; moreover, there have been instances lowering the semestral averages for mutability with large error bars in recent times evidencing short intervals indicating activity in a rather continuous way. In the case of zone B the recovery is still under way so it is too soon to say anything at this time. Generally speaking we can observe that mutability values were high and their error bars were small just before a major earthquake in zones A, B, and D.

Semestral averages for intervals between consecutive seisms and their corresponding error bars are very different among the different regions. Both values decrease during the aftershock regime but no clear trend could be found prior to a large earthquake.

As for the Calm zone C the mutability semestral averages are clearly increasing reaching 1.8 with narrowing error bars. Although each zone can have different thresholds for triggering of a major event, such value or slightly lower ones have been present just before large earthquakes in the other zones. Eventually zone C is showing a behavior that should be further studied at the expectation of future large quakes.

Let us close by answering the 5 points raised in the introduction thus summarizing previous discussions and conclusions. 1) Both Shannon entropy and mutability give similar responses to a major earthquake and its immediate aftershock period, however they are independent and non-correlated during the quieter periods. 2) Shannon entropy deals with the distribution as a whole while mutability and the entropy defined in natural time (which is dynamic and not static (Varotsos et al., 2007)) deal with a sequential distribution of intervals of natural time; this allows to the latter be more effective in providing larger contrasts if the values of the characteristics parameters. 3) The recovery time and background activity are very well characterized by mutability allowing to discriminate among different zones. 4) The mutability semestral averages reflect the seismic activity of the different zones indicating where the subduction is relatively fluent or where the process could be stuck. 5) A combined analysis points to zone A as stuck for many

years and zone C slowly decreasing fluency in the subduction process which can be indication for accumulation of energy in this zone.

This paper deals with the analysis of an important, but particular, seismic zone, namely the Nazca-South American subduction front. Our results show that the use of mutability and Shannon entropy may distinguish the different dynamics within this trench, and, specially, the fact that mutability may give a clue on the recovery time in a given region between major earthquakes. Certainly, further studies should be made in order to establish the general applicability of this approach, both by studying other seismic zones, and artificial catalogs, such as those given by the ETAS model. We expect to develop this in future publications.

**APPENDIX**

In this appendix we provide examples of the way mutability if calculated following Eq. (3) for time sequences similar to those found in this problem, using $\nu = 24$ as it is done dynamically in previous presentation. Each column in Table V lists one of these sequences representing intervals between consecutive seisms in minutes. First column, called "Even", is monotonic assigning one-hour interval evenly. Second column. called "Converging", is constructed by means of two intercalated sequences: one ascending and the other descending, so correlations are diluted. Third column, called Random, is formed by a randomly generated sequence. Fourth column, called Sequential, is formed by a monotonic increase of the intervals so it is highly correlated. As is can be readily found all columns average around 60 minutes between consecutive registers.

| i | Even | Converging | Random | Sequential |
|---|------|-----------|--------|-----------|
| 1 | 60 | 30 | 57 | 48 |
| 2 | 60 | 90 | 112 | 49 |
| 3 | 60 | 32 | 9 | 50 |
| 4 | 60 | 88 | 49 | 51 |
| 5 | 60 | 34 | 60 | 52 |
| 6 | 60 | 86 | 73 | 53 |
| 7 | 60 | 36 | 14 | 54 |
| 8 | 60 | 84 | 112 | 55 |
| 9 | 60 | 38 | 9 | 56 |
| 10 | 60 | 82 | 49 | 57 |
| 11 | 60 | 40 | 90 | 58 |
| 12 | 60 | 80 | 40 | 59 |
| 13 | 60 | 42 | 55 | 60 |
| 14 | 60 | 78 | 49 | 61 |
| 15 | 60 | 44 | 67 | 62 |
| 16 | 60 | 76 | 35 | 63 |
| 17 | 60 | 46 | 87 | 64 |
| 18 | 60 | 74 | 67 | 65 |
| 19 | 60 | 48 | 67 | 66 |
| 20 | 60 | 72 | 49 | 67 |
| 21 | 60 | 50 | 21 | 68 |
| 22 | 60 | 70 | 77 | 59 |
| 23 | 60 | 52 | 38 | 60 |
| 24 | 60 | 68 | 108 | 61 |
| $\mu$ | 0.1875 | 1.2347 | 1.5670 | 0.3854 |

TABLE V. Example of 4 time sequences (second to fifth columns) averaging 60 minutes between consecutive events. Mutability values for each column are given in the last row. The first column lists the sequence.

Results for the mutability of each column are given in the last row. As it could have been anticipated the Even sequence has the least information leading to the lowest mutability value. Next is Sequential, which reflects a monotonic increase in the time intervals. Notoriously higher is Converging where correlations are poor. The highest mutability value is for the Random sequence in spite a few values are repeated; if no repetitions are present and/or the interval span is higher the mutability value would be even larger.

It can be noticed that even in a 24-instant sequence mutability values can span an order of magnitude, This is even more so for real interevent sequences where intervals can reach several hours (a thousand minutes or more) thus differentiating behaviors of seismic activity.

**ACKNOWLEDGEMENTS**

One of us (EEV) is grateful to Fondecyt (Chile) under contract 1190036, and Center for the Development of Nanoscience and Nanotechnology (CEDENNA) funded by Conicyt (Chile) under contract AFB180001 for partial support. DP thanks Advanced Mining Technology Center (AMTC) and the Fondecyt grant 11160452. VM thanks Fondecyt projects 1161711, and 1201967. RAM acknowledges support from FAPDF (Brazil).

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
