# Peer review of "Measuring the seismic risk along the Nazca-Southamerican subduction front: Shannon entropy and mutability"

_Natural Hazards and Earth System Sciences, 2020_

## Referee Comment (RC1) · Anonymous Referee #1 · 19 May 2020

General Comments: In my previous interactive comment (https://www.nat-hazards-earth-syst-sci-discuss.net/nhess-2019-309/nhess-2019-309-RC1.pdf) I emphasized that "the obtained results [by Vogel et al.] are interesting but the presentation does not conform to the existing literature although it uses ideas earlier published by other researchers". In their interactive comment (https://www.nat-hazards-earth-syst-sci-discuss.net/nhess-2019-309/nhess-2019-309-AC1.pdf), Vogel et al. consent to my comments stating that "all of which have been taken into account in the present version of the paper. We explicitly mention natural time now in the text and in the figures; previous literature is quoted". In contrast to this consent, however, in the present version of the manuscript Vogel et al. only partially addressed my comments, as is evident

from the following example: Having a look on their current figures 3, 4, 5, and 6, their figure captions state that they plot their results versus the sequence of events without clarifying that these plots are versus "natural time" (sequential number of events) as they stated in their interactive comment. Moreover, none of the works that introduced natural time analysis in 2001, 2002 and the relevant book in 2011 (cf. there are the first three works that I recommended in my previous Interactive Comment, see also below) have been included in the list of References. These three works are the following:

[P. Varotsos, N. Sarlis, and E. Skordas, Spatiotemporal complexity aspects on the Interrelation between Seismic Electric Signals and seismicity, Practica of Athens Academy, 76, 294-321, 2001. Available from http://physlab.phys.uoa.gr/org/pdf/p3.pdf]

[P.A. Varotsos, N.V. Sarlis, and E.S. Skordas, Long-range correlations in the electric signals that precede rupture, Phys. Rev. E, 66, 011902 (7), 2002.]

[Varotsos P.A., Sarlis N.V. and Skordas E.S., Natural Time Analysis: The new view of time. Precursory Seismic Electric Signals, Earthquakes and other Complex Time Series (Springer-Verlag, Berlin Heidelberg) 2011]

Specific Comments: First, in line 49 the authors write: "we make use of the seismic sequence itself analyzing . . . consecutive seisms.". It should be completed as follows: "we make use of the seismic sequence itself, as in natural time analysis (see, e.g., [EPL 96, 59002 (2011)]), analyzing . . . consecutive seisms".

Second, in line 76, the authors write: "A entropy could be defined in natural time by . . .". This is not accurate; it should be corrected as follows: "An entropy has been defined in natural time [Phys. Rev. E 68, 031106 (2003)] -being dynamic and not static [Phys. Rev. E 70, 011106 (2004); Appl. Phys. Lett. 91, 064106 (2007)- by . . ..

Third, in line 98, the authors write: "wlzip results clearly increase several months prior to large earthquakes (Vogel et al., 2017)." It should be completed as follows since Vogel et al. in 2017 were aware of natural time analysis of seismicity results "wlzip

results clearly increase several months prior to large earthquakes (Vogel et al., 2017), thus being in accordance with natural time analysis which reveals [EPL 96, 59002 (2011)] that before major earthquakes there is a crucial time scale of around a few to several months in which changes in the correlation properties of physical quantities like seismicity or crustal deformation are observed"

Fourth, in line 155 the authors write: "However, in our case the value of each vector component is the interevent time itself, thus temporal information is still kept in the time series." In my previous interactive comment, I pointed out that one of the major applications in natural time analysis can be found in the paper [Appl. Phys. Lett. 91, 064106 (2007)] which unfortunately has not been included by the authors in the list of References. In this paper, which is just one example, the interevent time itself was used, and in particular the interevent time between consecutive heart beats is studied (see Fig.1 (a),(b) of this paper) in natural time analysis by computing the entropy change under time reversal. Hence, the aforementioned excerpt of the authors should either be deleted, or reworded for the sake of an accurate information od the readers as follows: "In our case, the value of each vector component is the interevent time itself, which has been also used in the natural time analysis of electrocardiograms by considering the interevent time between consecutive heartbeats [Appl. Phys. Lett. 91, 064106 (2007)]."

Fifth, in line 201, the authors write: "but also on the time sequence of the intervals while Shannon entropy depends only on the distribution." In view of my previous comment, the above excerpt should be reworded as follows: "but also on the time sequence of the intervals, which has been also used in the natural time analysis of the consecutive heartbeat intervals, while Shannon entropy depends only on the distribution [Appl. Phys. Lett. 91, 064106 (2007)]."

Sixth, concerning the lines 380-383 (which are the lines 333-335 of the previous version), I have already asked the authors (in my previous interactive comment) to mention natural time which makes clear the difference between Shannon entropy and mutability.

I suggest the following improvement by restoring the lines 333 and 334 of the earlier version and adding a few words concerning natural time in order to make clear the above difference. Thus, this improvement now reads: "The difference between Shannon entropy and mutability evidenced after the recovery time are due to the handling of a static distribution by the former while the latter considers the order in which registers entered in the distribution in accordance with the concept of natural time." The meaning of the above difference will become clear provided that the following two completions (seventh and eighth) will be available to the reader since they explicitly state that while Shannon entropy is static, the entropy in natural time is dynamic as explained in detail in the two references [Phys. Rev. E 71, 011110 (2005); Appl. Phys. Lett. 91, 064106 (2007)] that are unfortunately missing in the manuscript.

Seventh, in lines 297 and 298, the authors write: "Shannon entropy considers the visit to a state without considering the order in which these visits take place, ..." It should be completed as follows: "Shannon entropy considers the visit to a state without considering the order in which these visits take place (which is of paramount importance for the entropy in natural time being dynamic entropy and not a static one [Phys. Rev. E 71, 011110 (2005); Appl. Phys. Lett. 91, 064106 (2007)], ..."

Eighth, in lines 406 and 407 the authors write: "Shannon entropy deals with the distribution as a whole while mutability deals with a sequential distribution of intervals of natural time;". It should be completed as follows: "Shannon entropy deals with the distribution as a whole while mutability and the entropy defined in natural time (which is dynamic and not static [Appl. Phys. Lett. 91, 064106 (2007)]) deal with a sequential distribution of intervals of natural time;".

In short, the authors should proceed to the above-mentioned changes to frame their present findings in accordance with the existing literature before the acceptance of the manuscript for publication.
* * *
2020-86, 2020.

---

## Referee Comment (RC2) · Anonymous Referee #2 · 30 May 2020

I consider that the work can be published taking into account the following considerations:

The authors analyzed time series constructed with the elapsed times between earthquakes from seismic catalogues recorded on four zones of the Nazca plate. Their study is sustained by the analysis of two quantities, the Shannon entropy and the mutability introduced by Vogel et al., (2012). Concerning with the mutability, the authors do not describe neither properties nor advantages of the wlzip when its compressor is used to analyze time series, can they give information about the relationship with the numerical values of the wlzip and the underlying complexity in a time series? For instance,
if the time series is periodic, random or fractal, how the behavior of the compressor is in those cases? Regarding with the 4 chosen zones, why there were consider the epicenters of the largest earthquakes as center of each zone? (as is shown in the table I) Also, the latitudes why were selected with intervals of 4 degrees approximately? Is there some reason from a seismological point of view? The authors should explain why the zone between A and B was not considered in this analysis, I think that this region could be similar with the zone C in the sense that there is not a large earthquake during the study period.

From figure 1 and table I can be observed that Zone C is almost completely inserted within the zones B and D, so that the corresponding time series of zone C contains information of the respective B and D time series. The authors must clear how they can distinguish the joint information.

The authors wrote that the first bin in the histograms of A, B and D, in the figure 2, represent mainly the activity of the aftershocks after the larger earthquakes occurred within the respective zones. The authors must to specify the criterion that allows differentiate between the aftershocks (in terms of the position and time occurrence after a main shock) and the possible background seismicity. Also, the authors do not describe the number of earthquakes occurred during the analyzed period (2011-2017 and 2009-2017 for zone D), and not specify the number of events in their time series.

In the lower panel, In Figures 3-6, the abscissa labelled "Events" corresponds to the succession of filtered seisms identified by the same label i used to define ti, nevertheless, in order to clarify the analyzed period, the authors must specify the dates of the periods and the seisms (red stars) referred.

Regarding the comments around the figures 9-12 the authors claim: "The first comment here is evident: these 4 regions present different seismic behaviors so we have to discuss them separately". The authors only analyzed the inter-event times series, nevertheless, they have to explain clearly what do they understand seismic behavior,

because some other seismic parameters like the magnitude, or released energy which are not considered in this study.

In figures 9, 10 and 12 the authors should set up with red stars the dates that identify the date when each large earthquake occurred.

Within the text is cited (Vogel et al. (2017)) and, in the references section there are two papers of Vogel et al. (2017), the authors must to write Vogel et al. (2017a) and Vogel et al. (2017b) to avoid confusion.

---

## Author Comment (AC1) · 6 Jun 2020

Answer To Referee 1. Thank you for your detailed suggestions which we have followed closely in the present (third) version of our paper. In particular natural time is explicitly mentioned in the captions of figures 3, 4, 5 and 6. Each one of the 8 points you elaborated have been considered in the way you proposed and the corresponding references are quoted. We think the concept of natural time is now better included and discussed along mutability.

---

## Author Comment (AC2) · 6 Jun 2020

Referee 2 1.- Concerning with the mutability, the authors do not describe neither properties nor advantages of the wlzip when its compressor is used to analyze time series, can they give information about the relationship with the numerical values of the wlzip and the underlying complexity in a time series? For instance,if the time series is periodic, random or fractal, how the behavior of the compressoris in those cases?

The referee correctly pointed out that we did not present any detail concerning mutability and wlzip since we have presented it in a previous paper (Vogel et al., Tectonophysics, 2017a) and other quoted papers. In addition, a few examples are given in

the Appendix. However, we have now inserted a couple of sentences in the second paragraph of Subsection "Data recognizer" to provide a general description along the line indicated by the Referee. Moreover, the differences and advantages of mutability with respect to Shannon entropy are now discussed in several passages of the paper following the natural time discussions suggested by the other Referee.

2.- Regarding with the 4 chosen zones, why there were consider the epicenters of the largest earthquakes as center of each zone? (as is shown in the table I) Also, the latitudes why were selected with intervals of 4 degrees approximately? Is there some reason from a seismological point of view? The authors should explain why the zone between A and B was not considered in this analysis, I think that this region could be similar with the zone C in the sense that there is not a large earthquake during the study period.

The aim of this article is to analyze regions close to megathrust occurrence in Chile with the wlzip method. In order to meet this main aim we have used seismic data sets recorded close to the epicenter of three large earthquakes occurred in Chile (with moment magnitudes greater than 8.0). This was the reason to define zones A, B, and D. During the analysis it appeared interesting to include the area in between zones B and D since two important plate slides had occurred south and north of this area, to search for any behavior indicating the risk of future earthquakes. Thus, we took this zone without a megathrust in order to compare the results between the three large earthquakes and, as we called, a "calm period". We are aware of the overlap this implies but the main feature is that zone C is free of a seism of magnitude above 8.0, unlike the others. The choice of zones centered in the major earthquake has a geometrical reason, given our interest in studying the information close to such events. Regarding the choice of 4 degrees, it turned out to be a geographical span where enough data could be collected, so that good statistics is achieved. All of the subduction front is of interest, in particular the zone between A and B that you mentioned. However, we thought that only one "calm zone" close to two major earthquakes was more relevant for the purposes of the

present study. The fifth paragraph (new) in the "Data organization" section explains better the zone definition. Thank you for your comment and queries, which helped us to clarify this point.

3.- From figure 1 and table I can be observed that Zone C is almost completely inserted within the zones B and D, so that the corresponding time series of zone C contains information of the respective B and D time series. The authors must clear how they can distinguish the joint information.

This was answered in the previous query. As already said, a new paragraph was added to explain this point. In addition Table II deals with the times covered for each zone which allows us to appreciate that zone C is defined differently, in accordance with its special characteristics and purposes.

4.- The authors wrote that the first bin in the histograms of A, B and D, in the figure 2, represent mainly the activity of the aftershocks after the larger earthquakes occurred within the respective zones. The authors must specify the criterion that allows differentiate between the aftershocks (in terms of the position and time occurrence after a main shock) and the possible background seismicity. Also, the authors do not describe the number of earthquakes occurred during the analyzed period (2011-2017 and 2009-2017 for zone D), and not specify the number of events in their time series.

With respect to the statement about the large number of aftershock after the main earthquake of each zone we have made two comparisons: A) a monthly comparison before and after the main earthquake (possible in zones B and D only as explained in the paper); B) A comparison of the number of seisms for the full zone and the number of seisms for a restricted "square" of only two degrees in each direction around the main seism. This discussion is now included as the fifth paragraph of the "Data organization" subsection of the revised manuscript. As for the main earthquakes in each time series they are represented by stars in figures 3-6: The number of events in each time series was given (and still is) in the third paragraph of this same subsection: 6891, 6626,

2824, and 6356 for zones A, B, C, and D respectively.

5.- In the lower panel, In Figures 3-6, the abscissa labelled "Events" corresponds to the succession of filtered seisms identified by the same label i used to define ti, nevertheless, in order to clarify the analyzed period, the authors must specify the dates of the periods and the seisms (red stars) referred.

That was a bit complicated since the number of stars is a bit large during the aftershock regime. We have solved your query differently, but pointing to the same goal: we have prepared a Table (present Table II) to interpret in dates the milestones in the abscissa axis of the events panel.

6.- Regarding the comments around the figures 9-12 the authors claim: "The first comment here is evident: these 4 regions present different seismic behaviors so we have to discuss them separately". The authors only analyzed the inter-event times series, nevertheless, they have to explain clearly what do they understand seismic behavior, because some other seismic parameters like the magnitude, or released energy which are not considered in this study.

Yes, you are absolutely right. Except for filtering we do not deal with magnitude, and released energy is not considered here. We have rephrased this discussion completely. This can be found in the paragraph commenting figures 9-12 indicated by the referee.

7.- In figures 9, 10 and 12 the authors should set up with red stars the dates that identify the date when each large earthquake occurred.

Yes, it was done, and it really helps to recognize the semester with the largest activity. Thanks!

8.- Within the text is cited (Vogel et al. (2017)) and, in the references section there are two papers of Vogel et al. (2017), the authors must to write Vogel et al. (2017a) and Vogel et al. (2017b) to avoid confusion.

Sorry. Yes, it was confusing. It is now corrected. Thank you.

---

## Referee Comment (RC3) · Anonymous Referee #2 · 8 Jun 2020

In my opinion, the questions asked in the first review were fully answered, so I have no further comments and the manuscript can be published now.
* * *

---

## Author Comment (AC3) · 8 Jun 2020

I have tried to upload the revised version of the paper but I have not found the way of doing it. By e-mail I sent a copy to the editorial.

---

## Author Comment (AC4) · 19 Jun 2020

Thank you for your helpful suggestions. As already stated we considered all of your remarks and now the discussion is closed.

---

## Author Response (AR1)

July 16, 2020

Prof. Oded Katz

Editor

Natural Hazards and Earth System Sciences

REF: MS # nhess-2020-86

Dear Prof. Katz:

We are writing to you directly (with copy to the journal) due to some confusion concerning the correspondence mediated by the referees and the editorial system. Let me list the exchanges to date:

**Concerning Referee 1**

May 19: RC1 "Reviewer´s comments" a detailed list of suggestions indicating the lines where to improve the paper.

June 06: AC1 "Reply to Referee 1" where we point out that we followed the suggestions of the referee line by line; the answers were in the new text itself as stated in our Reply. Since it was not possible to upload the new version we sent it to the editorial office on this date; apparently this new version was not forwarded neither to the referee or to you.

June 08 AC3 "Revised version of the paper" we let the referee know that the new version of the paper including all corrections he/she suggested was available.

**Concerning Referee 2**

May 30: RC2 "Review" Several comments and suggestions by Referee 2.

June 06: AC2 "Answer to Referee 2". We tell the referee how we considered each one of the points in his/her review.

June 08: RC3: "Accept". Referee 2 accepts our response and says that "the article can be published now".

June 19: AC4: "Reply to Referee 2" we thank him/her for the valuable comments.

We send herewith the version of the new manuscript where we have highlighted the answers to each referee: in cyan aspects dealing with the detailed points raised by Referee 1 and in purple answers to the queries and comments of Referee 2. You will see that all comments of both referees have been fully considered. Answers to both referees are also included.

We hope you can help us to continue ahead to get the approval of our paper.

Best regards

Eugenio E. Vogel

Corresponding Author

MS # nhess-2020-86